# Endothelin-1 is associated with mortality that can be attenuated with high intensity statin therapy in patients with stable coronary artery disease

Ruizhu Lin[1], Juhani Junttila[1,2,3], Jarkko Piuhola[1], E. Samuli Lepojärvi[1], Johanna Magga[1,3], Antti M. Kiviniemi[1], Juha Perkiömäki[1], Heikki Huikuri[1], Olavi Ukkola[1], Mikko Tulppo[1] & Risto Kerkelä [1,2,3 ✉]

## Abstract

**Background** All coronary artery disease (CAD) patients do not benefit equally of secondary prevention. Individualized intensity of drug therapy is currently implemented in guidelines for CAD and diabetes. Novel biomarkers are needed to identify patient subgroups potentially benefitting from individual therapy. This study aimed to investigate endothelin-1 (ET-1) as a biomarker for increased risk of adverse events and to evaluate if medication could alleviate the risks in patients with high ET-1.

**Methods** A prospective observational cohort study ARTEMIS included 1946 patients with angiographically documented CAD. Blood samples and baseline data were collected at enrollment and the patients were followed for 11 years. Multivariable Cox regression was used to assess the association between circulating ET-1 level and all-cause mortality, cardiovascular (CV) death, non-CV death and sudden cardiac death (SCD).

**Results** Here we show an association of circulating ET-1 level with higher risk for all-cause mortality (HR: 2.06; 95% CI 1.5–2.83), CV death, non-CV death and SCD in patients with CAD. Importantly, high intensity statin therapy reduces the risk for all-cause mortality (adjusted HR: 0.05; 95% CI 0.01–0.38) and CV death (adjusted HR: 0.06; 95% CI 0.01–0.44) in patients with high ET-1, but not in patients with low ET-1. High intensity statin therapy does not associate with reduction of risk for non-CV death or SCD.

**Conclusions** Our data suggests a prognostic value for high circulating ET-1 in patients with stable CAD. High intensity statin therapy associates with reduction of risk for all-cause mortality and CV death in CAD patients with high ET-1.

## Plain language summary

Patients with coronary artery disease (CAD) – in which the blood vessels supplying the heart become blocked - need careful management to prevent adverse outcomes related to their disease, such as a heart attack or sudden cardiac death. Identification of markers in the blood to predict adverse outcomes would help to improve the care of patients with CAD. Here, we find that higher circulating levels of endothelin-1 (ET-1), a protein secreted normally to maintain blood pressure, associate with greater risk of death in CAD patients. Cholesterol-lowering statin therapy used at high intensity (high dosage) can counteract the increased risk of death observed in CAD patients with high ET-1. Therefore, circulating ET-1 level could be used as a marker to predict the risk of death in CAD patients, and an indication for high intensity statin therapy. Our findings could help clinicians to improve the management of patients with CAD.

[1] Research Unit of Biomedicine and Internal Medicine, University of Oulu, Oulu, Finland. [2] Medical Research Centre Oulu, Oulu University Hospital and University of Oulu, Oulu, Finland. [3] Biocenter Oulu, University of Oulu, Oulu, Finland. ✉email: Risto.Kerkela@oulu.fi

Secondary prevention among patients with coronary artery disease (CAD) has improved immensely during the last few decades. However, not all patients are benefitting equally from secondary prevention and accessible markers for identification of patients at higher risk are direly needed[1–4]. Several biomarkers have been studied in this respect, but a comprehensive risk assessment tool is lacking. While some patient groups in this population have been identified and different target levels and therapy regime have been implemented (e.g. diabetes in secondary prevention[3,5]), there is a need to construct a robust group of risk markers with enough power to guide the therapy among high-risk patients.

Endothelin-1 (ET-1), secreted primarily by endothelial cells and vascular smooth muscle cells, functions through $ET_A$ and $ET_B$ receptors and serves a role in maintaining basal level of vascular tone, contributing to systemic and pulmonary artery resistance. In addition to being a potent vasoconstrictor, ET-1 has pro-inflammatory and mitogenic effects promoting vascular smooth muscle growth, cardiac hypertrophy and myocardial fibrosis[6]. Selective $ET_A$ and dual $ET_A/ET_B$ receptor antagonists are in clinical use for treatment of primary pulmonary arterial hypertension[7] and $ET_A$ receptor blocker atrasentan has shown efficacy in reducing the risk of renal events in patients with diabetes and chronic kidney disease[8].

Prior studies have shown prognostic role for ET-1 in patients with acute myocardial infarction[9] as well as in patients with stable CADs[10]. In the present study, we identify an association of circulating ET-1 levels and mortality among patients with stable CAD in over 10-year follow-up. Additionally, we find that high intensity statin therapy holds the potential to offset the increased mortality associated with elevated ET-1.

## Methods

**Study protocol and population.** Patient blood samples were obtained from the ARTEMIS study (ClinicalTrials.gov identifier NCT01426685)[11,12], a prospective observation study. The ARTEMIS study was conducted at Oulu University Hospital's Division of Cardiology during 2007–2012, after enrollment visit, the follow-up was through phone call. The inclusion criteria for the study were angiographically documented CAD with or without T2D. CAD was defined as more than one vessel with >50% stenosis detected by coronary angiography, and the diagnosis of T2D was done according to the World Health Organization standard (fasting glucose level ≥7.0 mmol/L or 2-h post-load glucose level in the oral glucose tolerance test ≥11.1 mmol/L).

The exclusion criteria included left ventricular ejection fraction (LVEF) < 35%, NYHA class IV, pregnancy, life expectancy <1 year and end-stage renal failure requiring dialysis. Blood and echocardiographic parameters were collected at the enrollment visit, the time of which was at least 3 months after the coronary angiography or the last revascularization.

In total, there were 1946 patients in the present study, and ET-1 concentration was acquired from 1945 patients. During the follow-up (mean 7.86 ± 2.12 years), 218 participants died with 116 due to CV death and 102 for non-CV death. 50 out of 116 CV deaths were SCDs. The overall population consisted of 31.8% (N = 618) females, 42.8% (N = 833) participants with type 2 diabetes (T2D), with an average age of 67 years IQR [61, 73].

The patients were assigned to low, moderate or high intensity statin groups based on the daily intensity of statin therapy. The low intensity statin group consisted of patients who were on simvastatin 10 mg, pravastatin 10 mg, pravastatin 20 mg, fluvastatin 20 mg or fluvastatin 40 mg; the moderate intensity statin group included patients who took simvastatin 20 mg, simvastatin 30 mg, simvastatin 40 mg, rosuvastatin 5 mg, rosuvastatin 10 mg,

atorvastatin 10 mg, atorvastatin 20 mg, pravastatin 40 mg, pravastatin 80 mg, fluvastatin 80 mg or lovastatin 40 mg; and the high intensity statin group comprised of patients who were on atorvastatin 40 mg, atorvastatin 80 mg, rosuvastatin 20 mg or rosuvastatin 40 mg[13].

**Outcome measurement.** The primary endpoint was all-cause mortality. Secondary endpoints were cardiovascular (CV) death, non-CV death and sudden cardiac death (SCD). SCD was defined by a witnessed death within one hour of the onset of symptoms, or unwitnessed death where the patients was last seen alive and stable within 24-h before found deceased.

**Biomarker assay.** All laboratory tests were carried out with standardized methods after 12-h overnight fast. Biomarkers for inflammation, cardiac function and renal function were acquired from the blood and urine samples and analyzed in the Oulu University Hospital laboratory. ET-1 was measured from the serum samples by using Quantikine Endothelin-1 ELISA kit (catalog #DET100, R&D Systems).

The serum samples were obtained by allowing the blood to clot for 30 min at room temperature, followed by centrifugation at 2000 g for 10 min. The serum was then aliquoted and stored at −20 °C until analyzed. Samples had not been thawed prior the ET-1 level analysis. The interval between the sample collection and ET-1 analysis was 6–12 months.

The analyses for ET-1 levels were performed manually and in duplicates for each sample. According to manufacturer, the intra-assay precision for the assay is 1.9%–4.0% and the inter-assay precision is 5.3%–7.6%. The sensitivity of the assay, assessed by minimum detectable dose of ET-1, was from 0.031 to 0.207 pg/mL, with a mean of 0.087 pg/mL in 34 samples.

**Statistics.** To select the approach of description of scalar variables, normality test was conducted. If the scalar variable followed normality, values are expressed as mean ± SD and examined by Student's $t$ test. Otherwise, values are expressed as median [interquartile range (IQR)], assessed with non-parametric Wilcoxon rank-sum comparison. Categorical data is expressed as counts (percentage) and analyzed with chi-square ($\chi^2$) test. When expected frequency was below 5, Fisher-exact test was applied instead.

Receiver-operating characteristic (ROC)[14] curve analysis was conducted to obtain the optimal cutoff value of ET-1 to predict all-cause mortality with fitting into logistic regression. The optimal threshold was defined by maximizing the difference between true positive rate (sensitivity) and false negative rate (1-specificity), i.e., Youden index.

Univariable and multivariable Cox proportional hazard (PH) regression models were used to estimate hazard ratio and 95% confidence interval (95% CI) of variables. In model 1, traditional clinical risk factors age, sex, T2D, body mass index (BMI), systolic blood pressure (sysBP), and B-type natriuretic peptide (BNP) were included into the variable selection. In model 2, the following clinically relevant variables were assessed for adjustment: age, sex, T2D, BMI, sysBP, BNP, diastolic blood pressure (diaBP), smoking, prior MI, prior percutaneous coronary intervention (PCI) or coronary artery bypass grafting (CABG), and NYHA classification. Based on model 2, model 3 additionally included creatinine clearance, left ventricular ejection fraction (LVEF), percentage of glycated hemoglobin A1c (%GHbA1C), low-density lipoprotein (LDL), insulin, high-density lipoprotein (HDL), total cholesterol and triglyceride levels. To address possible confounding effect of other medications, the use of

statins was adjusted for age, sex, T2D, NYHA, ACEIs, ARBs, β-blockers, CCBs and diuretics (model medications).

In models 1–3 and model medication, a stepwise forward selection of candidate variables based on Akaike Information Criterion (forward-AIC)[15,16] was applied to acquire the best set of variables in each model. The AIC selected variables according to both fitness and complexity (over-fitting) of a model by introducing a penalty for each inclusion of variable. The single variable that reduced AIC the most was the first variable retained in the model. Addition of a variable to the model required the added variable together with previously retained variables to outperform the last-step retained variables. Harrell concordance index (C-index) was also calculated to assess the ability of the model to discriminate outcomes. Alternatively, in model S4, univariable CoxPH was used to calculate variable significance, and if a result of $p < 0.05$ was obtained, variable was included for multivariable adjustment. The set of variables used for selection was the same as in model 3.

The PH assumption was examined by inspecting log-minus-log (LML) plots against time for both high ET-1 and low ET-1 groups. If LML crossed and thus violated assumption, log-rank test was applied for survival analysis. Kaplan-Meier[17,18] survival curve was statistically evaluated by log-rank test. For quartile analysis, the relation of ET-1 level to the outcome was analyzed by categorizing quartiles as continuous variables. In model 3, collinearity was assessed by computing variance inflation factor[19], which was no greater than 2 for each independent variable.

The statistical significance was defined as $p < 0.05$, and all tests were 2-sided. For multiple comparisons between ET-1 level and risk of endpoint events, $p$-value $< 0.0125$ (0.05/4) was defined statistically significant for each of the four endpoints. In analysis for association of intensity of statin therapy with risk of all-cause mortality and CV death, the $p$-value $< 0.025$ (0.05/2) was defined statistically significant. Excel 2016 and SPSS 25 were used for data collection. Kaplan-Meier survival plot were constructed by lifelines module of Python, and all statistical comparison were performed with R version 4.0.4.

**Ethics.** The study protocol was approved by the ethics committee of the Northern Ostrobothnia Hospital district, and written informed consents were obtained from all the patients when recruited during 2007–2012. The study complied with the Declaration of Helsinki[11,20,21].

**Reporting summary.** Further information on research design is available in the Nature Portfolio Reporting Summary linked to this article.

## Results

**High circulating ET-1 associates with increased mortality.** In total there were 1946 patients in the study, and ET-1 concentration was acquired from 1945 patients, with a median concentration of 1.51 pg/mL (IQR 1.25–1.88 pg/mL). Area under the curve (AUC) from ROC analysis indicated that the accuracy derived from ET-1 as continuous variable to predict all-cause mortality was 0.684 (95% CI: 0.646–0.722), and the best threshold (maximum sensitivity and specificity) was 1.585 pg/mL (Fig. S1). The specificity achieved by ET-1 with a cut-off of 1.58 pg/mL was 0.593, the sensitivity was 0.711 and the AUC was 0.653 (95% CI: 0.621–0.686). 845 patients (43.4%) had a circulating ET-1 level higher than the cutoff (ET-1 > 1.58 pg/mL) and were designated as high ET-1 group, while 1100 patients with ET-1 ≤ 1.58 pg/mL were assigned to low ET-1 group.

Patients with high ET-1 levels were older (69 IQR [63, 75] *vs* 66 IQR [60, 72], $p < 0.0001$), were more often women (36.4% *vs*

28.2%, $p < 0.0001$) and had more frequently T2D (58.5% *vs* 30.8%, $p < 0.0001$) (Table 1). The analysis for LVEF showed no difference between low and high ET-1 groups, while heart rate, BMI, sysBP and BNP were elevated in high ET-1 group. Patients in high ET-1 group had increased %GHbA1C and insulin levels, and reduced creatinine clearance. The total cholesterol level and LDL showed no statistical difference between the high and low ET-1 groups.

Regarding medications, patients in high ET-1 group were more frequently treated with angiotensin-converting enzyme inhibitors (ACEIs), Angiotensin II receptor blockers (ARBs), diuretics and calcium channel blockers (CCBs). The use of lipid lowering agents or β-adrenergic blockers (β-blockers) showed no difference between high and low ET-1 groups.

When comparing to patients in low ET-1 group, patients with ET-1 > 1.58 had a significantly higher rate of all-cause mortality (18.3% vs 5.7%, $p < 0.0001$), CV death (9.6% vs 3.2%, $p < 0.0001$), non-CV death (8.6% vs 2.6%, $p < 0.0001$) and SCD (4.4% vs 1.2%, $p < 0.0001$) (Table 1). Kaplan-Meier survival curve was plotted to indicate disease outcome based on threshold of 1.58 pg/mL of circulating ET-1 concentration (Fig. 1).

In univariable analyses for all-cause mortality, CV death, non-CV death and SCD indicated that the risks in patients of ET-1 > 1.58 group were 3–4 times higher comparing to patients in low ET-1 group (Table 2). Multivariable adjustment was performed with 3 models using forward-AIC Cox regression (Methods).

In each of the three models, patients in high ET-1 group remained at significantly higher risk for all-cause mortality (model 1: HR: 2.22; 95% CI 1.63–3.02; $p < 0.001$; model 2: HR: 2.13; 95% CI 1.56–2.92; $p < 0.001$; model 3: HR: 2.06; 95% CI 1.5–2.83; $p < 0.001$), CV death, non-CV death and SCD (Table 2). The highest risk conferred by ET-1 > 1.58 pg/mL was the risk for SCD (Table 2). Despite more variables were tested from model 1 to model 3, the forward-AIC selection avoided overfitting and ensured model simplicity, while C-index increased slightly in each endpoint (all-cause mortality: 0.78 to 0.79; CV death: 0.80 to 0.81; non-CV death: 0.76 to 0.78; SCD: 0.75 to 0.78). ET-1 > 1.58 pg/mL retained in each of the four endpoints, indicating improved ability to predict the risk of death with the inclusion of ET-1. To note, for SCD, ET-1 was the first variable that entered each model, and the single inclusion of ET-1 accounted for 41.42% (model 3) of AIC reduction. The approach of variable selection for adjustment that was based on univariable significance (model S4, see Methods) is also included in Supplementary Data 1. The model S4 indicated high ET-1 associated risk for all-cause mortality (HR: 2.14; 95% CI 1.56–2.94; $p < 0.001$), CV death (HR: 1.8; 95% CI 1.17–2.78; $p = 0.008$), non-CV death (HR: 2.46; 95% CI 1.56–3.9; $p < 0.001$) and SCD (HR: 2.48; 95% CI 1.25–4.94; $p = 0.01$).

Regrouping the studied population by the rank of ET-1 concentration revealed that each quartile increase in ET-1 concentration was associated with an increase in all-cause mortality, rate of CV death, non-CV death and SCD (Fig. 2, Table S1). Baseline characteristics across ET-1 quartiles are shown in Table S1. Unadjusted risk for all-cause mortality, CV death, non-CV death and SCD increased through the four ET-1 quartiles and robustly correlated with ET-1 levels (trend $p$-value $< 0.001$ for each cause of death, Table 3). After adjustment (model 3), the increase of risk conferred by ET-1 followed linearity for all-cause mortality (trend $p < 0.001$) and non-CV death (trend $p < 0.001$), but not for CV death and SCD. Patients belonging to the fourth quartile (Q4) of ET-1 concentration held significantly higher risk for all-cause mortality and non-CV death (HR: 2.01; 95% CI 1.26–3.23; $p = 0.004$ and HR: 2.63; 95% CI 1.39–4.98; $p = 0.003$, respectively) (Table 3).

**Table 1 Baseline characteristics of patients.**

|  | ET1 = < 1.58 | ET1 > 1.58 | p |
| --- | --- | --- | --- |
| N | 1100 | 845 |  |
| Age (year) | 66 [60, 72] | 69 [63, 75] | <0.0001 |
| T2D (%) | 339 (30.8) | 494 (58.5) | <0.0001 |
| Sex (%) (female) | 310 (28.2) | 308 (36.4) | 0.0001 |
| LVEF (%) | 65.60 [60.30, 70.10] | 65.40 [58.82, 70.38] | 0.3076 |
| BMI (kg/m²) | 27.00 [25.00, 30.00] | 29.00 [26.00, 32.00] | <0.0001 |
| sysBP (mm Hg) | 144 [128, 159] | 147 [133, 163] | <0.0001 |
| diaBP (mm Hg) | 76 [70, 83] | 77 [70, 85] | 0.1149 |
| BNP (pg/mL) | 3.76 [3.09, 4.38] | 4.08 [3.37, 4.84] | <0.0001 |
| Creatinine clearance (mL/min) | 89.66 [73.55, 112.98] | 85.03 [66.34, 110.26] | 0.0005 |
| Insulin (mIU / L) | 10.50 [6.90, 16.10] | 13.65 [8.07, 21.42] | <0.0001 |
| GHbA1C (%) | 6.00 [5.70, 6.40] | 6.30 [5.90, 7.10] | <0.0001 |
| Total cholesterol (mmol/L) | 3.80 [3.40, 4.40] | 3.90 [3.40, 4.50] | 0.1692 |
| HDL (mmol/L) | 1.24 [1.04, 1.46] | 1.20 [1.01, 1.41] | 0.0066 |
| LDL (mmol/L) | 2.10 [1.80, 2.60] | 2.20 [1.80, 2.60] | 0.3899 |
| Triglyceride (mmol/L) | 1.13 [0.85, 1.56] | 1.32 [0.98, 1.80] | <0.0001 |
| Resting heart rate (beats/min) | 58 [52, 64] | 62.00 [55, 69] | <0.0001 |
| NYHA classification (%) |  |  |  |
| NYHA Class I | 395 (36.3) | 458 (54.6) | <0.0001 |
| NYHA Class II | 33 (3.0) | 36 (4.3) |  |
| Medication (%) |  |  |  |
| ARBs | 286 (26.0) | 272 (32.2) | 0.0034 |
| ACEIs | 420 (38.2) | 363 (43.0) | 0.0368 |
| Lipid-lowering therapy | 1012 (92.1) | 765 (90.5) | 0.2592 |
| Diuretics | 285 (25.9) | 381 (45.1) | <0.0001 |
| β-blockers | 960 (87.4) | 744 (88.0) | 0.6948 |
| CCBs | 213 (19.4) | 261 (30.9) | <0.0001 |
| History (%) |  |  |  |
| Prior MI | 527 (48.0) | 402 (47.6) | 0.9046 |
| Prior PCI or CABG | 876 (80.0) | 668 (79.1) | 0.6851 |
| Smoking | 67 (6.1) | 100 (11.9) | <0.0001 |
| Death (%) |  |  |  |
| All-cause of Death | 63 (5.7) | 155 (18.3) | <0.0001 |
| Cardiovascular Death | 35 (3.2) | 81 (9.6) | <0.0001 |
| non-Cardiovascular Death | 29 (2.6) | 73 (8.6) | <0.0001 |
| Sudden cardiac death | 13 (1.2) | 37 (4.4) | <0.0001 |

Values were expressed as mean ± SD, median [IQR], or counts (%) as appropriate. T2D, type 2 diabetes; *LVEF* left ventricular ejection fraction, *BMI* body mass index, *sysBP* systolic blood pressure, *diaBP* diastolic blood pressure, *BNP* B-type natriuretic peptide, *%GHbA1C* percentage of glycated hemoglobin A1c, *HDL* high-density lipoprotein, *LDL* low-density lipoprotein, *NYHA* New York Heart Association classification, *ARBs* Angiotensin II receptor blocker, *ACEIs* Angiotensin-Converting Enzyme Inhibitors; *β-blockers*, β-adrenergic blockers, *CCBs* calcium channel blockers, *MI* myocardial infarction, *CABG* coronary artery bypass surgery, *PCI* percutaneous coronary intervention. ET-1, endothelin-1.

**High intensity statin therapy attenuates high ET-1 associated risks.** We further investigated for a potential medical therapy that could offset the increased mortality associated with high circulating ET-1 levels. Kaplan-Meier survival curve (Fig. S2) revealed that only lipid-lowering agents held the potential to reduce incidence of primary endpoint in patients with high ET-1, but not ARBs, ACEIs, β-blockers, CCBs or diuretics. Given that the medication for lowering lipids varied, we focused on selected patients who received statins only ($N = 1397$) and re-assigned them to low, moderate or high intensity statin groups based on the intensity of therapy[13]. The use of statins was compared to those who did not use lipid-lowering agents ($N = 828$ vs $N = 87$ in the low ET-1 group; $N = 569$ vs $N = 80$ in the high ET-1 group). The basic characteristics of selected patients in each group are shown in Table S2. Violin plots of ET-1 and LDL levels in the eight groups are shown in Fig. S3. Comparison of each on-statin group to respective no statin group indicated no difference in ET-1 levels, and no difference in ET-1 levels was observed between various intensities of statins (Fig. S3). LDL levels were significantly lower with statin therapy, but the reduction was not dependent on the intensity of statin therapy.

Within high ET-1 group, log-rank test (Fig. 3) showed that the statins associated with lower all-cause mortality (chi-square = 9.6,

$p = 0.02$) and CV death (chi-square = 10.5, $p = 0.01$), but not with non-CV death or SCD. However, such effect on all-cause mortality and CV death was not observed in low ET-1 group (Fig. S4). For high ET-1 group, univariable analysis (Table 4) supported that high intensity statin therapy associated with markedly decreased risk for all-cause mortality (HR: 0.13; 95% CI: 0.03–0.58; $p = 0.007$). Patients in both moderate statin (HR: 0.46; 95% CI 0.25–0.85; $p = 0.013$) and high statin (HR: 0.17; 95% CI 0.04–0.73; $p = 0.018$) groups had reduced risk for CV death. After multivariable adjustment (Table 4; model 3), in high ET-1 group, high intensity statin exhibited robust attenuation of risk for all-cause mortality (HR: 0.05; 95% CI 0.01–0.38; $p = 0.004$) and CV death (HR: 0.06; 95% CI 0.01–0.44; $p = 0.006$) compared to those who did not take statins. The risk reduction for CV death in moderate statin subgroup was 62% (HR: 0.38; 95% CI 0.2–0.71; $p = 0.002$) (Fig. 3, Table 4), but was not sufficient to mitigate the risk for all-cause mortality after adjustments. Patients with low statin subgroup did not show a decrease in risk for all-cause mortality or CV death. Adjustment using method in model S4 was also tested for the effect of statins to reduce the risk of all-cause mortality and CV death, and those analysis were returned with similar data as shown in Supplementary Data 2.

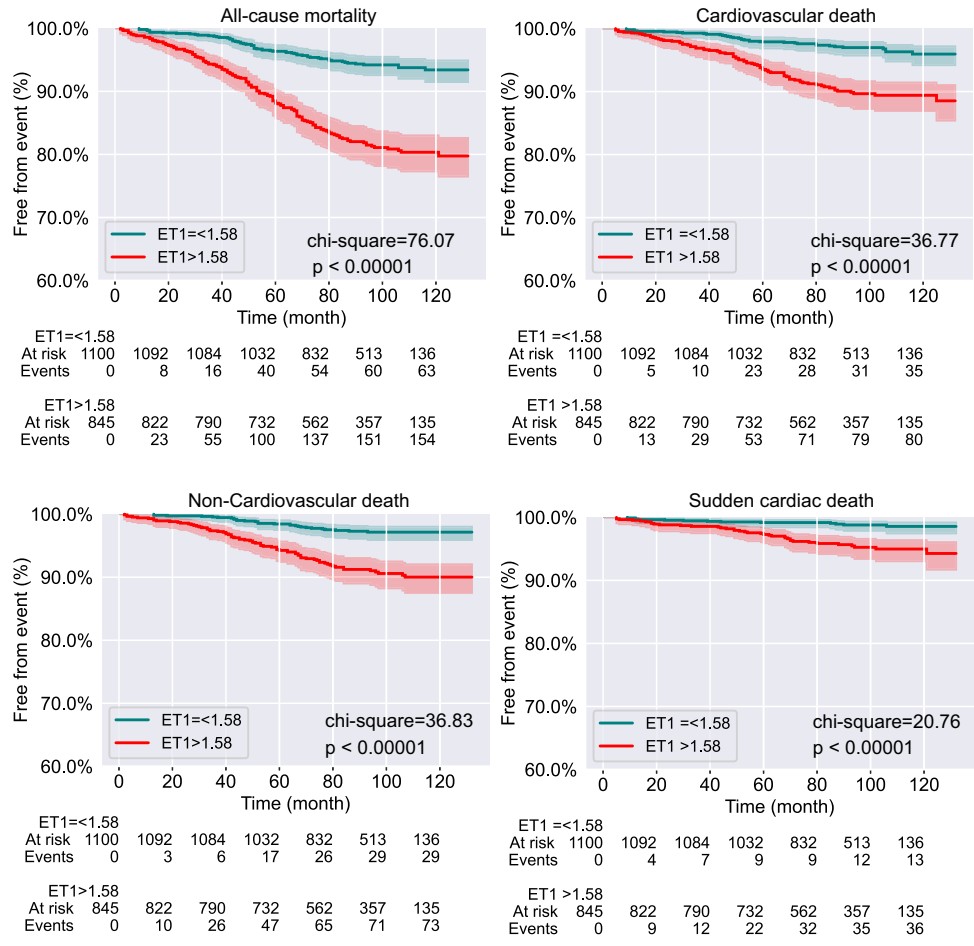

**Fig. 1 Kaplan-Meier curves for primary and secondary outcome in patients with low and high circulating ET-1.** Shown are freedom from all-cause mortality, and rate of freedom from cardiovascular death, non-cardiovascular death and sudden cardiac death over follow up time. Chi-square and *p* value were derived from log-rank test between low and high ET-1 groups. The shadows along each line indicated confident interval.

Multivariable adjustment with model medications (see Methods) showed that the reduction of risk conferred by the use of high and/or moderate intensity statins was independent from the use of ACEIs, ARBs, β-blockers, CCBs or diuretics (Supplementary Data 3).

## Discussion

Our data suggest a role for ET-1 in prognosis of patients with stable coronary artery disease. In addition, the increased risk for cardiovascular mortality associated with high ET-1 could be decreased with high intensity statin therapy. These results have potential implications for therapy among CAD patients at high risk even when receiving apparently optimal medical management. Individualized therapy in secondary prevention has been implemented in guidelines already regarding CAD patients with diabetes, but more subgroups with detailed therapy suggestions are needed[3, 5]. Our current findings suggest that measuring ET-1 plasma concentration could be a tool for a risk assessment of coronary artery disease patients.

Previously, a study from Sabatine et al. (Sabatine study) has shown a role for ET-1 in predicting CV death and heart failure in patients with stable CAD[10]. Our data supports previous findings but also extends to non-CV death and SCD. Interestingly, after adjustment, in each model, ET-1 associated risk for non-CV death is higher and more robust than the risk for CV death. This indicates that non-CV death, rather than CV death, is the

primary risk of death associated with high ET-1. The strongest risk derived from high ET-1 is risk for SCD.

ET-1 has been regarded the most potent vasoconstrictor identified in human[22,23] and also functions as a key regulator of contractility and hypertrophic response in the left ventricle[24,25]. Dysregulation of ET-1 has been observed in patients with symptomatic atrial fibrillation, and restoration of sinus rhythm after pulmonary vein isolation was associated with reduced ET-1 level in these patients[26]. Interestingly, despite being an essential vasoconstrictor, ET-1 can also induce arrhythmia irrespective to myocardial ischemia[27,28]. Therefore, ET-1 may act as an intrinsic arrhythmogenic factor with a distinct electrophysiological properties[27]. ET-1 associated arrhythmias show no change on conduction time[27], but may be attributed to its potential to prolong the cardiac action potential duration[29]. Several mechanisms have been proposed for ET-1 related arrhythmogenic action, including altered $Ca^{2+}$ mobilization and inward $K^+$ current[29].

T2D is a risk factor for CV diseases and it's incidence has been rising between 1996–2016[30,31] The Sabatine study included 16.2% of patients with T2D[10], whereas the present study, designed to study and identify risk markers between CAD patients with and without T2D, constitutes 42% of population with T2D. Given that T2D is essentially a microvascular disease, the role of ET-1, a potent vasoconstrictor, may be overestimated in the present study. However, after adjustment with four different models, ET-1 remained an independent risk factor regardless of T2D. The

**Table 2 Unadjusted and multivariable-adjusted association of ET-1 level and clinical outcomes.**

| | Hazard Ratio | *p*-value | C-index | −ΔAIC | −ΔAIC by ET-1 (%) |
|---|---|---|---|---|---|
| **All-cause mortality** | | | | | |
| Univariable (*N* = 1945) | 3.41 (2.54–4.56) | <0.001 | - | - | - |
| Model 1 (*N* = 1945) | 2.22 (1.63–3.02) | <0.001 | 0.78 | −239.25 | −46.61 (19.48%) |
| Model 2 (*N* = 1918) | 2.13 (1.56–2.92) | <0.001 | 0.79 | −241.12 | −46.07 (19.11%) |
| Model 3 (*N* = 1887) | 2.06 (1.5–2.83) | <0.001 | 0.79 | −242.24 | −45.09 (18.61%) |
| **CV death** | | | | | |
| Univariable (*N* = 1945) | 3.19 (2.15–4.75) | <0.001 | - | - | - |
| Model 1 (*N* = 1945) | 1.84 (1.21–2.8) | 0.005 | 0.8 | −146.38 | −6.88 (4.7%) |
| Model 2(*N* = 1918) | 1.74 (1.14–2.67) | 0.011 | 0.81 | −151.20 | −6.82 (4.51%) |
| Model 3 (*N* = 1887) | 1.84 (1.2–2.83) | 0.005 | 0.81 | −156.98 | −7.12 (4.53%) |
| **non-CV death** | | | | | |
| Univariable (*N* = 1945) | 3.49 (2.27–5.36) | <0.001 | - | - | - |
| Model 1 (*N* = 1945) | 2.69 (1.71–4.21) | <0.001 | 0.76 | −89.67 | −21.47 (23.95%) |
| Model 2 (*N* = 1918) | 2.54 (1.61–3.99) | <0.001 | 0.77 | −89.81 | −21.3 (23.71%) |
| Model 3 (*N* = 1887) | 2.56 (1.63–4.02) | <0.001 | 0.78 | −92.93 | −19.51 (21%) |
| **SCD** | | | | | |
| Univariable (*N* = 1945) | 3.9 (2.07–7.35) | <0.001 | - | - | - |
| Model 1 (*N* = 1945) | 2.7 (1.39–5.24) | 0.003 | 0.75 | −40.93 | −18.82 (45.98%) |
| Model 2 (*N* = 1918) | 2.7 (1.36–5.37) | 0.005 | 0.77 | −41.05 | −19.14 (46.63%) |
| Model 3 (*N* = 1887) | 2.58 (1.3–5.11) | 0.007 | 0.78 | −46.0731 | −19.08 (41.42%) |

*AIC* Akaike Information Criterion, −ΔAIC absolute reduction of AIC when ET-1 entering model,
% −ΔAIC the percentage of AIC reduction by ET-1 inclusion in terms of total reduction of AIC in each model.
In model 1, traditional clinical risk factors of age, sex, T2D, body mass index (BMI), systolic blood pressure (sysBP), and B-type natriuretic peptide (BNP) were included into variable selection.
In model 2, the following clinically relevant variables were assessed for adjustment: age, sex, T2D, BMI, sysBP, BNP, diastolic blood pressure (diaBP), smoking, prior myocardial infarction (MI), prior percutaneous coronary intervention (PCI) or coronary artery bypass grafting (CABG), and NYHA classification.
Based on model 2, model 3 (as well as model S4) further included circulating creatinine clearance, LVEF, percentage of glycated hemoglobin A1c (%GHbA1C), low-density lipoprotein (LDL), insulin, high-density lipoprotein (HDL), total cholesterol and triglyceride level.
*ET-1* endothelin-1, *CV death* cardiovascular death, *non-CV death* non-cardiovascular death, *SCD* sudden cardiac death.

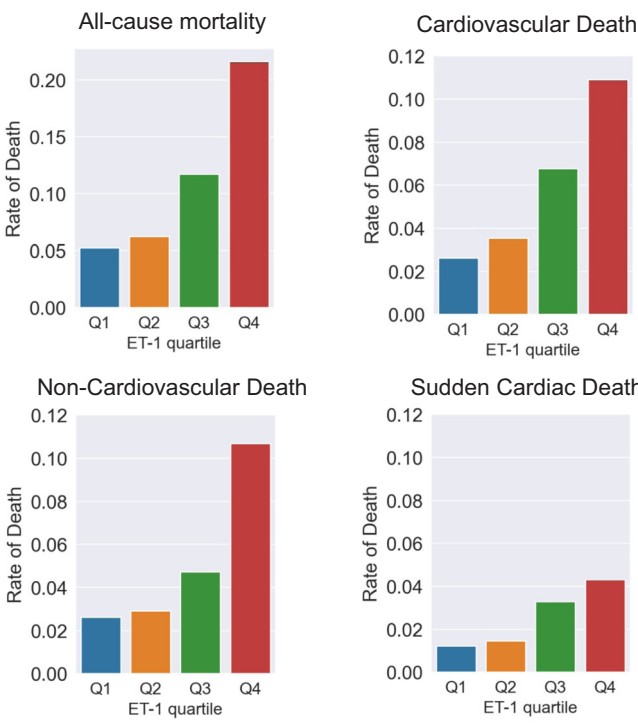

**Fig. 2 Incidence rates of endpoints according to quartiles of ET-1.** Patients were regrouped according to four quartiles of ET-1 concentration. Q1, first quartile of ET-1 concentration; Q2, second quartile of ET-1 concentration; Q3, third quartile of ET-1 concentration; Q4, fourth quartile of ET-1 concentration. All-cause mortality, rate of cardiovascular death, non-cardiovascular death and sudden cardiac death in each ET-1 quartile.

current findings thus support the prognostic value for high circulating ET-1 also in CAD patients with T2D. Of note, the majority of population in both Sabatine study and our study consisted of male patients (81.1% and 68.2%, respectively). The sex imbalance in both studies indicates a need for research with focus on females to address women's health.

The current study and previous study[10] both show prognostic value for high ET-1 in patients with CAD, while a recent study did not find predictive value for ET-1 in the development of CAD[32]. Analysis for circulating ET-1 concentration did not associate with the extent of CAD and plaque phenotype in stable chest pain patients without a history of CAD but with an intermediate pre-test probability of obstructive CAD[32].

The statins have shown remarkable efficacy in reducing LDL, a well-established risk factor for atherosclerotic cardiovascular disease, and in both primary and secondary prevention of cardiovascular events[33,34]. The relative risk reduction with statins across populations is proportional to the absolute reduction in LDL, and the greatest absolute benefit is achieved in patients with the highest baseline risk[35]. Apart from high LDL, dysglycemia and other lipoprotein abnormalities, relatively little is known of other biological factors that may affect the benefit of an individual patient from statin therapy.

Interestingly, we find that high intensity statin therapy reduces the high ET-1 associated risk for all-cause mortality and CV death, but not risk for non-CV death or SCD. Therefore, the risk reduction in all-cause mortality by high intensity statin therapy could logically mainly stem from decreased risk for CV death. Sabatine study showed that ACEI treatment reduces the risk for CV death or heart failure by approximately 40% in patients in highest ET-1 quartile[10]. In the current study cohort, 25–45% of patients were on ACEI/ARB therapy and moderate to high intensity statin therapy was observed to reduce the risk for CV death in high ET-1 group by more than 50% independently from ACEI/ARB therapy. Importantly, our data show that treatment of patients with high intensity of statins reduces the risk for all-cause mortality and CV death in high ET-1 group but not in low ET-1 group. Patients with a higher absolute risk are likely to benefit the most from any risk reduction, including statin therapy[35]. On the other hand, the majority of vascular events occur in patients at lower risk as they are much larger in numbers, and an absolute LDL reduction (mmol/l) results in at least as large relative risk

**Table 3 Relation of ET-1 level and disease outcomes.**

| | Univariable analysis | | | Multivariable analysis (model 3) | | |
|---|---|---|---|---|---|---|
| | Hazard Ratio | *p*-value | Trend *p* | Hazard Ratio | *p*-value | Trend *p* |
| All-cause mortality | | | | | | |
| ET-1 (Q2) | 1.19 (0.71–2.02) | 0.507 | <0.001 | 0.95 (0.55–1.61) | 0.838 | <0.001 |
| ET-1 (Q3) | 2.29 (1.44–3.65) | <0.001 | | 1.37 (0.84–2.23) | 0.201 | |
| ET-1 (Q4) | 4.45 (2.89–6.83) | <0.001 | | 2.01 (1.26–3.23) | 0.004 | |
| CV death[a] | | | | | | |
| ET-1 (Q2) | 1.36 (0.66–2.79) | 0.408 | <0.001 | 1.01 (0.48–2.14) | 0.979 | 0.056 |
| ET-1 (Q3) | 2.65 (1.4–5.04) | 0.003 | | 1.43 (0.72–2.85) | 0.302 | |
| ET-1 (Q4) | 4.47 (2.44–8.2) | <0.001 | | 1.49 (0.74–2.98) | 0.260 | |
| non-CV death | | | | | | |
| ET-1 (Q2) | 1.11 (0.52–2.37) | 0.781 | <0.001 | 0.93 (0.44–1.98) | 0.848 | <0.001 |
| ET-1 (Q3) | 1.85 (0.94–3.65) | 0.076 | | 1.26 (0.63–2.52) | 0.513 | |
| ET-1 (Q4) | 4.41 (2.4–8.1) | <0.001 | | 2.63 (1.39–4.98) | 0.003 | |
| SCD[a] | | | | | | |
| ET-1 (Q2) | 1.21 (0.41–3.61) | 0.729 | <0.001 | 0.99 (0.31–3.2) | 0.989 | 0.106 |
| ET-1 (Q3) | 2.78 (1.09–7.1) | 0.033 | | 1.91 (0.68–5.4) | 0.223 | |
| ET-1 (Q4) | 3.81 (1.54–9.45) | 0.004 | | 1.67 (0.56–4.93) | 0.356 | |

[a]For CV death and SCD, ET-1 quartile (as categorical variable) was not able to enter model 3 with forward-AIC selection or enter model S4. Therefore, shown are multivariable adjustment with all variables in model 3 but entering model together. The 1st quartile of ET-1 (Q1) was used as reference. *ET-1* endothelin-1, *CV death* cardiovascular death, *non-CV death* non-cardiovascular death, *SCD* sudden cardiac death

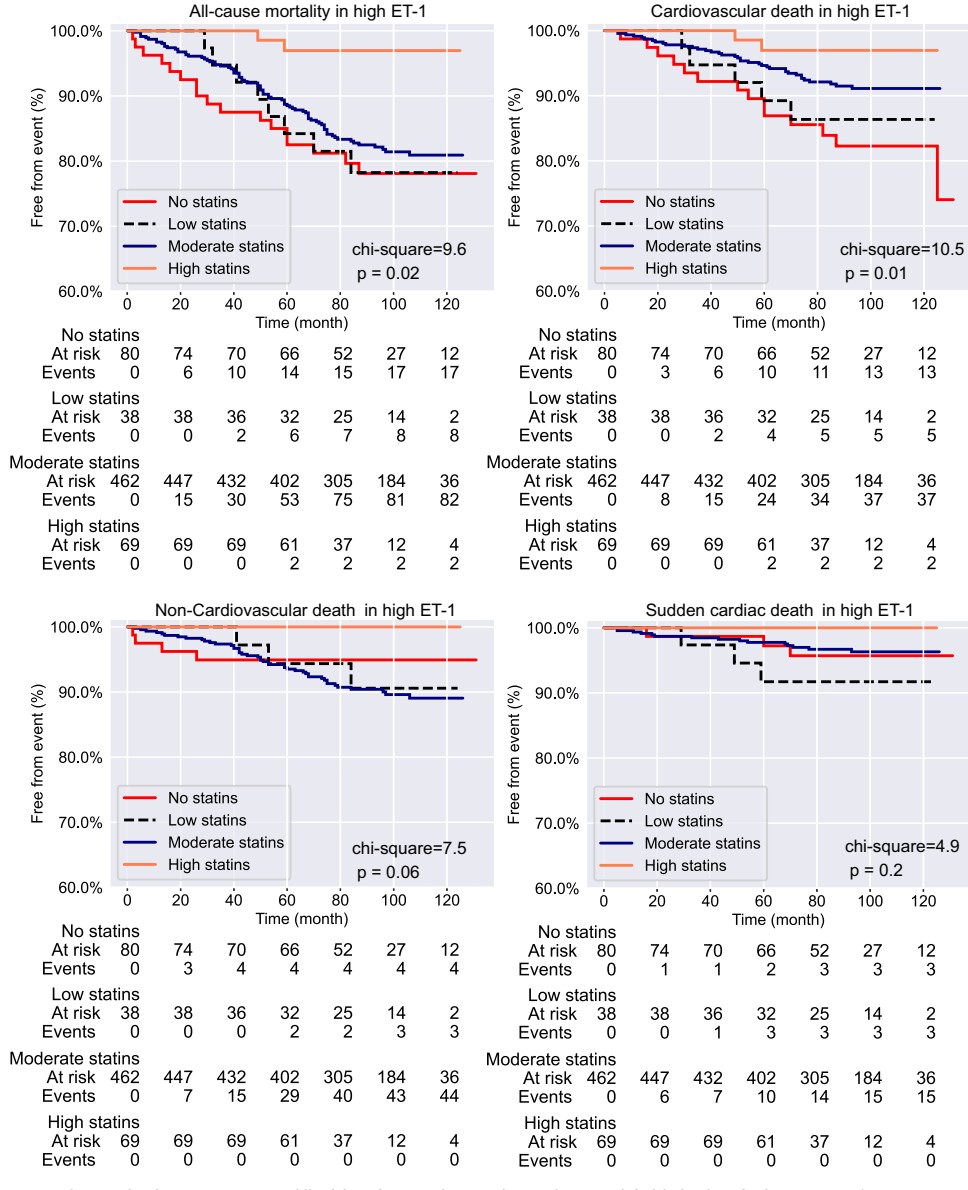

**Fig. 3 Kaplan-Meier curves for endpoint events stratified by the statin use in patients with high circulating ET-1.** Shown are rate of freedom from all-cause mortality, cardiovascular death, non-cardiovascular death and sudden cardiac death over follow-up time.

**Table 4 Statins and risk for all-cause mortality and CV death in high ET-1 group.**

| High ET-1 | Univariable analysis | | Multivariable analysis (model 3) | |
|---|---|---|---|---|
| | Hazard Ratio | *p*-value | Hazard Ratio | *p*-value |
| All-cause mortality | | | | |
| no statins | reference | | reference | |
| statins (low) | 0.96 (0.41–2.23) | 0.926 | 0.78 (0.33–1.82) | 0.567 |
| statins (moderate) | 0.8 (0.48–1.36) | 0.415 | 0.76 (0.45–1.3) | 0.32 |
| statins (high) | 0.13 (0.03–0.58) | 0.007 | 0.05 (0.01–0.38) | 0.004 |
| CV Death | | | | |
| no statins | reference | | reference | |
| statins (low) | 0.76 (0.27–2.12) | 0.603 | 0.54 (0.19–1.52) | 0.241 |
| statins (moderate) | 0.46 (0.25–0.85) | 0.013 | 0.38 (0.2–0.71) | 0.002 |
| statins (high) | 0.17 (0.04–0.73) | 0.018 | 0.06 (0.01–0.44) | 0.006 |

Low intensity statin group consist of simvastatin 10 mg, pravastatin 10 mg, pravastatin 20 mg, fluvastatin 20 mg and fluvastatin 40 mg; Moderate intensity statin group contains simvastatin 20 mg, simvastatin 30 mg, simvastatin 40 mg, rosuvastatin 5 mg, rosuvastatin 10 mg, atorvastatin 10 mg, atorvastatin 20 mg, pravastatin 40 mg, pravastatin 80 mg, fluvastatin 80 mg and lovastatin 40 mg; High intensity statin group comprises atorvastatin 40 mg, atorvastatin 80 mg, rosuvastatin 20 mg and rosuvastatin 40 mg. *ET-1* endothelin-1, *LDL* low-density lipoprotein, *CV death* cardiovascular death.

reduction of major vascular events in low-risk individuals[36]. Data from the current study suggests that among these patients at high risk, high circulating ET-1 levels can be used to stratify patients who will benefit the most from the statin therapy. We further find that the benefit is only associated with moderate and high intensity statin therapy. Additional studies are needed to investigate if the beneficial effect of statin therapy in patients with high ET-1 remains in follow-up of patients treated with statins according to the current LDL-C goals.

Prior meta-analysis indicates that statins can reduce the levels of circulating ET-1 irrespective of intensity and duration[37]. In the present study, the risk for all-cause mortality and CV death was attenuated by high intensity statin treatment in high-ET-1 group but not in low ET-1 group. However, circulating ET-1 levels were not altered by statins irrespective of dosage. Therefore, it is unlikely that protection conferred by statins in patients with high ET-1 is due to decrease in ET-1 levels. We assigned the patients to statin groups according to the potential of each statin to lower the LDL levels. While the patients on statin therapy had lower LDL levels than patients without lipid-lowering therapy, the LDL levels in patients on statins were similar irrespective of the intensity of the therapy. It is therefore likely that an additional mechanism other than LDL lowering contributes to the protective effect of statins in patients with high ET-1. Prior study has shown that statins can inhibit ET-1 -induced vascular contraction and DNA synthesis in vascular smooth muscle cells, which might partially explain the beneficial effects in patients with high circulating ET-1[38]. LDL-independent anti-oxidation, anti-inflammatory effect, increased vaso-relaxation through enhancing NO bioavailability and possible regulation of endothelial function may also account for salutary effect of statins[39,40]. Besides, given that statins are a class of HMG-CoA reductase inhibitors, therefore all intermediate products along mevalonate pathway may also be affected by treatment with statins, which can produce an effect not caused but associated with LDL reduction.

As an observational study, despite that we attempted to rule out important confounders with adjustments, there might be confounders beyond observation or being excluded for adjustment due to selection, which may affect the association identified here.

The population in this study is well defined, but unfortunately there are no measurements of cholesterol level or ET-1 level or information on the intensity of statin treatment at follow-up. Therefore, it requires cautious interpretation of the observed association of statin therapy with attenuated risk. Moreover, while we observed association of high ET-1 with increased risk of mortality, the current data does not allow to determine if high ET-1 was also a cause for higher risk of death, which would

suggest potential for ET-1 receptor antagonists to reduce mortality in CAD patients with high ET-1.

Data from this observational cohort study shows that increased ET-1 concentration is associated with increased risk for all-cause mortality, CV death, non-CV death and SCD. Analysis of ET-1 levels may thus have prognostic value and be of benefit in identifying patients that should be subjected to more intensive therapy. Moreover, our data shows that high intensity statin therapy can substantially reduce the risk for all-cause mortality and cardiovascular death in stable CAD patients with high circulating ET-1. Given that the present study was hypothesis generating, independent studies with adequate power and randomization are warranted to validate the findings.

### Data availability
Source data for Fig. 1 and Fig. 3 in the main text with statistical analyses are provided in the table under each figure, and source data of Fig. 2 is provided in Table S1. The datasets analyzed in the current study and all other data are available from the corresponding author upon reasonable request.

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

## Acknowledgements

The authors thank Päivi Kastell for technical support. This work was supported by Academy of Finland [grant number 333349 to J.J. and grant number 297094 and 333284 to R.K.]; Finnish Foundation for Cardiovascular Research [to R.L., J.J., H.H., M.T. and R.K.]; by Jane and Aatos Erkko Foundation [to J.J., H.H. and R.K.] and the Sigrid Juselius Foundation (to J.J., H.H. and R.K.) and Business Finland [to M.T.].

## Author contributions

R.L. contributed to conceptualization, investigation, statistical analysis, data processing, data visualization, originally drafting, review and editing the manuscript. J.J. contributed to conceptualization, investigation, funding support, supervision, review and editing the manuscript. J.M. contributed to data validation, supervision, review and editing the manuscript. J.P. and O.U. contributed to review and editing the manuscript. E.S.L. and A.M. Kiviniemi contributed to data curation, data collection, review and editing the manuscript. J.P. contributed to data collection and curation. H.H. contributed to funding, supervision, review and editing the manuscript. M.T. contributed to funding, review and editing the manuscript. R.K. contributed to conceptualization, investigation, funding, supervision, data validation, drafting, review and editing the manuscript.

## Competing interests

The authors declare no competing interests.
