## [Peer Review File · Communications Medicine]

Reviewers' comments:

Reviewer #1 (Remarks to the Author):

With great interest, I have read this manuscript from Lin and colleagues reporting on the prognostic relevance of circulating endothelin-1 levels and the effects of statins in patients with chronic coronary syndrome.

Comments:

Lines 61-63: the authors put forth several statements, but do not back this up by references.

Line 78: the authors abbreviate myocardial infarction but don't use the abbreviation anymore throughout the manuscript.

In the methods, the authors should start by describing the ARTEMIS study and inclusion criteria before proceeding to report on the number of included patients.

Lines 101-107: The way these lines are constructed suggests that the patients in the groups took all of the aforementioned statins. Please rewrite.

Lines 108-109: Did the authors also obtain ethical approval for this study when they obtained informed consent? Or was this study later added? Did the ethical committee approve? Was this a prespecified analysis and if so, please provide a reference. If not, add this to the limitations.

Line 125-128: Could the authors replicate the intra-assay precision reported by the manufacturer? Do they have reference to this intra-assay precision?

Line 130: What kind of normality test was performed?

Section following line 235: The finding that high intensity statin therapy attenuates high ET-1 associated risks seems to be the most novel finding, yet all the figures are provided as supplemental figures.

Lines 265 – 267: This sentence seems grammatically incorrect.

Discussion section: The discussion about the study by Sabatine seems to lean towards an almost competitive element, where the number of included patients are compared. I would argue the authors should discuss how their findings are complementary to the study by Sabatine.

Discussion section: The authors discuss the role of ET-1 in arrhythmia, but in their findings, statin therapy does not reduce sudden cardiac death in fact suggesting that the benefit of statin therapy for reducing ET-1 and potentially thereby preventing events, is not through an arrhythmic pathway.

Discussion section: The authors omitted a recent study by Jukema RA et al. 2022 in Atherosclerosis.

Limitations: the authors should highlight that the data is observational and their findings are hypothesis generating. In fact, I think this should also be highlighted in the abstract and the conclusion.

Furthermore, could the authors comment in the discussion section on the use of ET-1 antagonists to prevent cardiovascular events? If indeed cardiovascular events are reduced by statin therapy, at least in part, through ET-1 attenuating properties, then ET-1 antagonist therapy in patients with high ET-1 levels, should also reduce cardiovascular events to prove causality between ET-1 attenuating therapy and preventing events.

Reviewer #2 (Remarks to the Author):

In this manuscript, Lin and colleagues investigate the association between increasing levels of ET-1 and fatal outcomes in a cohort of patients with significant obstructive disease on coronary angiography, who were followed for a mean in excess of 7 years. They report worsening outcomes with higher baseline ET-1 levels, which remain present after a number of adjustment models. In a subsequent analysis they report that the risk associated with higher ET-1 levels is reduced in patients treated with higher intensity statin therapy. The findings are interesting and talk to the concept of greater modifiable risk. The authors should consider a number of points.

1. As acknowledged, the literature already has reports of a relationship between ET-1 and adverse outcomes in patients with CAD. Accordingly, while there is some incremental information provided here, the novelty is limited.
2. While the authors have performed a number of adjustment models, the reality of residual confounding cannot be denied. There are substantial differences in these patients, they are likely to be different groups.
3. If I'm correct, the statin data reflects use at baseline? Given the long follow up, what was done to account for potential differences in doses and cessation?
4. I'm not sure I see any real clinical implication here. In the time that these patients have been followed there have been substantial changes to guidelines and LDL-C targets on the basis of large clinical trials. I'm not sure the current findings have much relevance.
5. I think it would be helpful to see how ET-1 net reclassify risk, rather than just provide AUC models.

RE: COMMSMED-22-0414

We thank reviewers and the editors for careful and constructive comments. Please find below point-by-point response to the referees' comments..

Reviewers' comments:

Reviewer #1 (Remarks to the Author):

With great interest, I have read this manuscript from Lin and colleagues reporting on the prognostic relevance of circulating endothelin-1 levels and the effects of statins in patients with chronic coronary syndrome.

Comments:

Lines 61-63: the authors put forth several statements, but do not back this up by references.

RE: We have added the following references:

1. Fox, K. A. A., Metra, M., Morais, J. & Atar, D. The myth of 'stable' coronary artery disease. *Nat Rev Cardiol* **17**, 9–21 (2020).
2. Fox, K. A. A. *et al.* Anti-thrombotic options for secondary prevention in patients with chronic atherosclerotic vascular disease: what does COMPASS add? *European Heart Journal* **40**, 1466–1471 (2019).
3. Cosentino, F. *et al.* 2019 ESC Guidelines on diabetes, pre-diabetes, and cardiovascular diseases developed in collaboration with the EASD. *European Heart Journal* **41**, 255–323 (2020).

Line 78: the authors abbreviate myocardial infarction but don't use the abbreviation anymore throughout the manuscript.

RE: We have removed the abbreviation.

In the methods, the authors should start by describing the ARTEMIS study and inclusion criteria before proceeding to report on the number of included patients.

RE: The first three paragraphs of the Method section have been rearranged.

Lines 101-107: The way these lines are constructed suggests that the patients in the groups took all of the aforementioned statins. Please rewrite.

RE: The section has been revised.

Lines 108-109: Did the authors also obtain ethical approval for this study when they obtained informed consent? Or was this study later added? Did the ethical committee approve? Was this a prespecified analysis and if so, please provide a reference. If not, add this to the limitations.

RE: The study protocol of ARTEMIS study was first approved by the local committee of research ethics of the Northern Ostrobothnia Hospital District. After that, the ARTEMIS study was conducted at Oulu University Hospital's Division of Cardiology during 2007-2012. A consecutive series of patients were recruited according to the pre-defined criteria in the protocol with written informed consent collected.¹⁻³ We have now added this information to the Ethics section of the Methods (lines 176-178).

The ethical permit allows for collection of blood samples for analysis, but it was not required at the time for the specific analyses to be listed in the permit. Analysis of ET-1 was one of the ~30 planned

analyses performed from the blood samples, and the analysis for ET-1 levels was performed within 6-12 months after sample collection. The study thus adheres with the ethical approval for the study.

References:

1. Junttila, M. J. et al. Type 2 diabetes and coronary artery disease: Preserved ejection fraction and sudden cardiac death. *Heart Rhythm* 15, 1450–1456 (2018).
2. Kenttä, T. et al. Effects of Exercise Rehabilitation on Cardiac Electrical Instability Assessed by T-Wave Alternans During Ambulatory Electrocardiogram Monitoring in Coronary Artery Disease Patients Without and With Diabetes Mellitus. *The American Journal of Cardiology* 114, 832–837 (2014).
3. Lepojärvi, E. S. et al. Usefulness of Highly Sensitive Troponin as a Predictor of Short-Term Outcome in Patients With Diabetes Mellitus and Stable Coronary Artery Disease (from the ARTEMIS Study). *The American Journal of Cardiology* 117, 515–521 (2016).

Line 125-128: Could the authors replicate the intra-assay precision reported by the manufacturer? Do they have reference to this intra-assay precision?

RE: The manufacturer of the kit provides intra-assay precision (3 samples measured 20 times) for the assay (https://www.rndsystems.com/cn/products/endothelin-1-quantikine-elisakit_det100), and their coefficients of variation (CV)% range is 1.9% - 4.0%.

To replicate the manufacturer assessment of intra-assay CV%, we randomly chose three plates where 40 samples were measured in duplicates. The average intra-assay CV% for the three plates were 1.28%, 2.52% and 4.61%. Our data is thus in accordance with the data from the manufacturer.

As our analysis was not designed for assessment of especially the intra-assay CV (2 vs 20 replicates of the same sample in one plate), we included the values from the manufacturer into the manuscript.

Line 130: What kind of normality test was performed?

RE: The total number of participants in the present study is 1946 (<2000), and R was used for statistical analysis. Therefore, Shapiro-Wilk test were applied for normality test for continuous variables.¹

Reference:

1. Yap, B. W. & Sim, C. H. Comparisons of various types of normality tests. *Journal of Statistical Computation and Simulation* 81, 2141–2155 (2011).

Section following line 235: The finding that high intensity statin therapy attenuates high ET-1 associated risks seems to be the most novel finding, yet all the figures are provided as supplemental figures.

RE: One out of the three figures regarding statin therapy is presented in the Figures (Fig. 3). Data in Figure 3 shows Kaplan-Meier curves for endpoint events stratified by the statin use in patients with high circulating ET-1.

In supplementary data, we present data on ET-1 and LDL concentrations according to statin use (Figure S3) and Kaplan-Meier curves for endpoint events in low ET-1 group according to statin use (Figure S4).

Lines 265 – 267: This sentence seems grammatically incorrect.

RE: The sentence has been revised.

Discussion section: The discussion about the study by Sabatine seems to lean towards an almost competitive element, where the number of included patients are compared. I would argue the authors should discuss how their findings are complementary to the study by Sabatine.

RE: We apologize for the tone of the discussion on the data from the Sabatine study and the discussion has been now modified. Please see Discussion, lines 280-285 and 296-305.

Discussion section: The authors discuss the role of ET-1 in arrhythmia, but in their findings, statin therapy does not reduce sudden cardiac death in fact suggesting that the benefit of statin therapy for reducing ET-1 and potentially thereby preventing events, is not through an arrhythmic pathway.

RE: Although statin therapy did not decrease sudden cardiac death (SCD), the strongest risk associated with high ET-1 was risk for SCD (Table 2) in the present study. Importantly, high ET-1 was the first variable selected entering each model of SCD among all clinically relevant variables, indicating a pivotal role of high ET-1 in predicting the risk of SCD.

Therefore, we feel it is necessary to discuss the role of ET-1 in arrhythmia.

Discussion section: The authors omitted a recent study by Jukema RA et al. 2022 in Atherosclerosis.

RE: This is an excellent suggestion. We have discussed the findings of the Jukema study in the Discussion, lines 306-310.

Limitations: the authors should highlight that the data is observational and their findings are hypothesis generating. In fact, I think this should also be highlighted in the abstract and the conclusion.

RE: The nature of the study is now also noted in the Abstract (first sentence of Methods section) and the Conclusions (lines 370 and 375-377).

Furthermore, could the authors comment in the discussion section on the use of ET-1 antagonists to prevent cardiovascular events? If indeed cardiovascular events are reduced by statin therapy, at least in part, through ET-1 attenuating properties, then ET-1 antagonist therapy in patients with high ET-1 levels, should also reduce cardiovascular events to prove causality between ET-1 attenuating therapy and preventing events.

RE: Data in the current study shows that statin therapy does not affect ET-1 levels (Fig. S3). ET-1 has multiple biological effects and statin therapy likely only affects a part of those. Endothelin-1, for example, enhances oxidative stress¹, whereas statins have been shown to attenuate oxidative stress in vascular endothelium.²

Concerning endothelin receptor antagonists (ERAs), a review³ (Supplement table S1) summarizes the clinical trials with ERAs in CAD patients across 30 years. In the initial years, the application of non-selective ERAs (tezosentan) did not show benefit in reducing mortality in patients with acute decompensated heart failure associated with acute coronary syndrome.^{4,5} Recent trial indicated a favorable effect for bosentan (non-selective ERA) in preventing major cardiovascular adverse events in Hispanic population with incipient peripheral arterial disease (PAD).⁶ A selective ET-A receptor blocker, BQ-123, administered in patients with ST elevation myocardial infarction immediately prior to primary percutaneous coronary intervention, also resulted in longer event-free survival (cardiovascular re-hospitalization) over a median of three-year follow-up.⁷ Therefore, types (selective vs non-selective) of ERAs, protocol of administration, targeting population and subgroup of CAD cohorts, all might affect the clinical readouts of ERAs.

We have added a notion on the potential of ET-1 receptor antagonists in lines 364-366.

References:

1. Dong, F. et al. Endothelin-1 enhances oxidative stress, cell proliferation and reduces apoptosis in human umbilical vein endothelial cells: role of ETB receptor, NADPH oxidase and caveolin-1. *Br J Pharmacol.* 2005 Jun; 145(3): 323–333.
2. Margaritis, M et al. Statins as Regulators of Redox State in the Vascular Endothelium: Beyond Lipid Lowering. *Antioxid Redox Signal.* 2014 Mar 10; 20(8): 1198–1215.
3. Barton, M. & Yanagisawa, M. Endothelin: 30 Years From Discovery to Therapy. *Hypertension* 74, 1232–1265 (2019).
4. O'Connor, C. M. et al. Tezosentan in patients with acute heart failure and acute coronary syndromes: Results of the randomized intravenous tezosentan study (ritz-4). *Journal of the American College of Cardiology* 41, 1452–1457 (2003).
5. Kaluski, E. et al. RITZ-5: randomized intravenous TeZosentan (an endothelin-A/B antagonist) for the treatment of pulmonary edema: A prospective, multicenter, double-blind, placebo-controlled study. *Journal of the American College of Cardiology* 41, 204–210 (2003).
6. De Haro, J., Bleda, S., Gonzalez-Hidalgo, C., Michel, I. & Acin, F. Long-Term Effects of Bosentan on Cardiovascular Events in Hispanic Patients with Intermittent Claudication: Four-Year Follow-up of the CLAU Trial: The CLAU Randomized Trial Long-Term Outcome. *Am J Cardiovasc Drugs* 19, 203–209 (2019).
7. Adlbrecht, C. et al. Peri-interventional endothelin-A receptor blockade improves long-term outcome in patients with ST-elevation acute myocardial infarction. *Thromb Haemost* 112, 176–182 (2014).

Reviewer #2 (Remarks to the Author):

In this manuscript, Lin and colleagues investigate the association between increasing levels of ET-1 and fatal outcomes in a cohort of patients with significant obstructive disease on coronary angiography, who were followed for a mean in excess of 7 years. They report worsening outcomes with higher baseline ET-1 levels, which remain present after a number of adjustment models. In a subsequent analysis they report that the risk associated with higher ET-1 levels is reduced in patients treated with higher intensity statin therapy. The findings are interesting and talk to the concept of greater modifiable risk. The authors should consider a number of points.

1. As acknowledged, the literature already has reports of a relationship between ET-1 and adverse outcomes in patients with CAD. Accordingly, while there is some incremental information provided here, the novelty is limited.

RE: Thank you for the comment. The association of high ET-1 with adverse outcomes only formed the first part of our conclusions. Importantly, we then examined for possible treatments that could counteract high ET-1 associated risks. When analyzing for potential of statin therapy in alleviating high ET-1 associated risks, we found that high and moderate intensity statin therapy have potential in alleviating the high ET-1 associated mortality.

Pubmed search (performed on February 4th 2023) for keywords “endothelin” and “statin” present in title or abstract of articles, and article type restricted into clinical trial, meta-analysis or randomized controlled trial but excluding reviews, systemic reviews, books and documents yielded 16 articles. None of those investigated the direct effect of statins on high ET-1 associated mortality. Thus, our second conclusion that only high and/or moderate intensity of statins possessed the therapeutic potential in antagonizing high ET-1 associated risks of all-cause death and CV death, establish the first evidence for potential of statins in reducing high ET-1 related risk for increased mortality.

2. While the authors have performed a number of adjustment models, the reality of residual confounding cannot be denied. There are substantial differences in these patients, they are likely to be different groups.

RE: The present study is a prospective observational cohort study (OBS), and residual confounding is one of the inherent disadvantages of OBS, owing to no pre-randomization. As the reviewer noticed that all adjustments, ranging from traditional clinical risk factors to clinically relevant variables from model 1 to model 3, were attempts to address the confounder issues. Unfortunately, due to the nature of the study, we still cannot fully eliminate residual confounding that beyond observation or being overlooked without selection.

The residual confounder issue is now noted in the Limitations (lines 357-359).

3. If I'm correct, the statin data reflects use at baseline? Given the long follow up, what was done to account for potential differences in doses and cessation?

RE: The statin data only reflected baseline use. As an observational cohort study, we are only able to provide novelty in establishing an association between treatments (or risk factors) and outcomes. It requires a well-designed clinical trial with careful monitoring to prove the effect of statins on high ET-1 associated risk of death, which is out of the scope of current study. The lack of follow-up on statin treatment has been addressed in the Limitations (line 360-363).

4. I'm not sure I see any real clinical implication here. In the time that these patients have been followed there have been substantial changes to guidelines and LDL-C targets on the basis of large clinical trials. I'm not sure the current findings have much relevance.

RE: The lower LDL-C targets in CAD patients in the current guidelines results in a higher number of patients being treated with moderate or high intensity statins. The current findings suggest that patients with high ET-1 should be treated with high intensity of statins irrespective of LDL-C level. It is thus likely that there remain patients that would benefit from high intensity statin therapy even though they have reached the target LDL-C level with a low or moderate dose of statins. Naturally, it needs to be investigated if the beneficial effect of statin therapy in patients with high ET-1 remains in follow-up of patients treated with statins according to the current LDL-C goals. We have added this notion to the Discussion (lines 334-336).

In Europe wide survey it has been previously determined that most patients with CAD have suboptimal lipid lowering therapy.¹ Thus, irrespective of LDL-C target levels, there are a number of patients that would potentially benefit from statin therapy. Among those there are also likely patients with high ET-1, who would particularly benefit from the statin therapy.

Reference:

1. De Backer, G. et al. Management of dyslipidaemia in patients with coronary heart disease: Results from the ESC-EORP EUROASPIRE V survey in 27 countries. *Atherosclerosis*. 2019 Jun;285:135-146. doi: 10.1016/j.atherosclerosis.2019.03.014

5. I think it would be helpful to see how ET-1 net reclassify risk, rather than just provide AUC models.

RE: Net reclassification index (NRI) is sometimes used for evaluating the improvement in prediction performance gained by adding a marker to a set of baseline predictors. However, the statistical properties of this measure are controversial, and several articles suggest caution in using the NRI as the basis for marker evaluation (for example, see¹⁻³)

Concerning the current data, inclusion of ET-1 ≥ 1.58 pg/mL to model 1 resulted in 16.7% improvement in NRI in prediction of risk for all-cause death (95% CI: 0.003 - 0.37). The NRI for prediction of CV death, non-CV death or SCD did not reach significance (CV death NRI: -0.03; 95% CI: -0.07-0.25; non-CV death NRI 15%, 95% CI: -0.03-0.33 and SCD NRI 1%, 95% CI: -0.01-0.18).

References:

1. Hilden J, Gerds TA. A note on the evaluation of novel biomarkers: do not rely on integrated discrimination improvement and net reclassification index. *Stat Med*. 2014 Aug 30;33(19):3405-14.
2. Kerr KF. Net Reclassification Index Statistics Do Not Help Assess New Risk Models. *Radiology*. 2023 Mar;306(3):e222343.
3. Pepe MS et al. The Net Reclassification Index (NRI): a Misleading Measure of Prediction Improvement Even with Independent Test Data Sets. *Stat Biosci*. 2015 Oct 1;7(2):282-295.

REVIEWERS' COMMENTS:

Reviewer #1 (Remarks to the Author):

My comments have been adequately addressed by the authors.

Reviewer #2 (Remarks to the Author):

The authors are thanked for their comments and revisions. I continue to have concerns with regard to the clinical implications of this manuscript. While the findings are interesting, they simply reflect observational data and the authors have extended a long extrapolation of baseline statin use to impact of what is essentially high modifiable risk.

RE: COMMSMED-22-0414B

We thank the editors and the reviewers for careful evaluation of our manuscript. Please find below point-by-point response to the Reviewers' comments.

Reviewer #1 (Remarks to the Author):

My comments have been adequately addressed by the authors.

RE: Thank you for positive comments.

Reviewer #2 (Remarks to the Author):

The authors are thanked for their comments and revisions. I continue to have concerns with regard to the clinical implications of this manuscript. While the findings are interesting, they simply reflect observational data and the authors have extended a long extrapolation of baseline statin use to impact of what is essentially high modifiable risk.

RE: It will indeed be of interest to investigate if the beneficial effect of high intensity statin therapy in patients with high ET-1 can be validated in a prospective study with adequate power and randomization.